# Analytical Pyrolysis of the Fungal Melanins from *Ochroconis* spp. Isolated from Lascaux Cave, France

Cesareo Saiz-Jimenez [1,*] , Pedro M. Martin-Sanchez [1,2] , Jose A. Gonzalez-Perez [1] and Bernardo Hermosin [1]

[1] Instituto de Recursos Naturales y Agrobiologia, IRNAS-CSIC, 41012 Sevilla, Spain; p.m.martin-sanchez@ibv.uio.no (P.M.M.-S.); jag@irnase.csic.es (J.A.G.-P.); hermosin@irnase.csic.es (B.H.)

[2] Section for Genetics and Evolutionary Biology (Evogene), Department of Biosciences, University of Oslo, 0316 Oslo, Norway

[*] Correspondence: saiz@irnase.csic.es

**Abstract:** Analytical pyrolysis is a tool widely used for investigating the nature of macromolecules. This technique has been applied with success for solving the chemical structure of geo- and biopolymers. However, with complex macromolecules, such as fungal melanins, analytical pyrolysis presents some drawbacks due to the fact that they rarely can be obtained in a pure grade and contains a mixture of polysaccharides, chitin, proteins, nucleic acids, and lipids, among other materials. Analytical pyrolysis permits the characterization of the compounds co-extracted or linked to the melanin. Nevertheless, our data revealed that under the conditions usually employed in the pyrolysis of melanins, the technique presented serious limitations due to the complex structure and diversity of components that suffer extensive thermal degradation through secondary reactions with the production of considerable number of artifacts. To the light of pyrolysis data, the melanin of *Ochroconis* spp. is not based on 3,4-dihydroxyphenylalanine (DOPA) or 1,8-dihydroxynaphthalene (DHN), and it is suggested that it could be a type of pyomelanin, based on the polymerization of homogentisic acid and other phenolic compounds.

**Keywords:** analytical pyrolysis; fungal melanins; *Ochroconis lascauxensis*; Lascaux Cave; *Eurotium echinulatum*; *Cryptococcus neoformans*; DOPA melanin; DHN melanin; pyomelanin

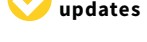



## 1. Introduction

In the last decade, Lascaux Cave in France has received considerable attention from media and scientists [1–7]. The interest in the biodeterioration processes that affect the cave lies in different fungal outbreaks that have affected its halls and passages (walls, ceiling, and Paleolithic paintings) since 2001. The most recent one was the formation of black stains on both limestone-rock walls and clayey sediments with a critical increase from 2006 to 2009. There were two new species of the fungal genus *Ochroconis*, *O. lascauxensis* and *O. anomala*, which produced intracellular melanins, isolated from the cave walls [6], but *O. lascauxensis* was the main contributor to the black stains on limestone rocks and their dissemination in the cave [5].

Structural investigation on fungal melanins is a challenging task because of their unknown macromolecular structure. Melanins can be found in microorganisms, plants and animals. Nicolaus [8] distinguished three types of melanins: eumelanin (skin, hair, feathers, melanoma, etc.), pheomelanin (red hair, feathers, foxes red fur) and allomelanin (plants, bacteria, and fungi). Classical works on melanins considered that these compounds are derived from diphenols, such as 3,4-dihydroxyphenylalanine (DOPA), 5,6-dihydroxyindole, catechol or 1,8-dihydroxynaphthalene (DHN), which polymerizes to form a macromolecule [8,9].

There were two more types of melanins: neuromelanin, a mixture of pheomelanin and eumelanin, formed from dopamine and 5-S-cysteinyldopamine, and pyomelanin, derived from homogentisic acid, recently added to the classical well-known melanins [10].

The nature and origin of the fungal melanins have been a topic extensively investigated in the last five decades with different results and conclusions [11–15]. In fact, for Eisenman and Casadevall [16] fungi may synthesize melanin from endogenous substrate via a DHN intermediate and, alternatively, some fungi produce melanin from DOPA. However, the detailed chemical structure of these melanins is unknown. Other authors stated that fungi often produced extracellular and intracellular melanins in culture media by polymerization of different phenols [17] and anthraquinones [18].

Some research programs have been conducted in Lascaux Cave aiming at disclosing the origin, nature, and evolution of the fungal outbreaks, and the colonization and dissemination of black stains produced by the activity of the *Ochoroconis* spp. [6,7]. Among the efforts to control fungal colonization, one of the targets is to know the type of melanin synthesized by the fungi and how to prevent their growth [6,19].

Several analytical approaches have been used to address melanin composition. They include different spectroscopies, which provide insights into the nature of the chemical functions [19], and chemical degradations which aim at releasing molecular information through cleavage of the macromolecule [8,12].

Previous studies characterized the melanins of *O. lascauxensis* and *O. anomala*, isolated from the cave, and one closely related reference species, *O. tshawytschae*, by both nuclear magnetic resonance (NMR) spectroscopy and surface-enhanced Raman spectroscopy (SERS), suggesting that *Ochoroconis* melanins are not related to the precursors typically attributed to the chemical structure of fungal melanins: DOPA or 5,6-dihydroxyindole. In addition, no clear evidence of the DHN precursor was found [6,19].

An alternative method for the study of melanins is analytical pyrolysis. This technique presents advantages versus chemical degradation due to the very small amount of sample needed, and the high sensitivity, and specificity. In this study, we analyzed the thermal decomposition products of the melanins extracted from *Ochroconis*. The melanins were pyrolyzed using different protocols and temperatures and the pyrolysis products separated by gas chromatography and identified by mass spectrometry. To obtain additional information, a thermochemolysis (thermally assisted hydrolysis and methylation) method was applied. The aim of the study was to characterize the chemical structure of the melanin isolated from *Ochroconis* using pyrolysis-gas chromatography-mass spectrometry in search of molecular markers able to provide information on melanin precursors.

## 2. Materials and Methods

This study is focused on the fungal melanins extracted from the mycelia of three type strains: *O. lascauxensis* CBS131815 (=LX A1), *O. anomala* CBS131816 (=LX CH40), and *O. tshawytschae* CBS100438. The protocols for melanin extractions, purification and acid hydrolysis were thoroughly described by De la Rosa et al. [19] as well as their elemental analyses and spectroscopic characteristics.

For analytical pyrolysis, the melanin samples (1 mg) were deposited on small crucible capsules (Eco-cup, Frontier Labs., Fukushima, Japan). Pyrolysis-gas chromatography-mass spectrometry (Py-GC-MS) was performed under inert helium atmosphere using a double-shot micro-furnace pyrolyzer (Frontier Labs. model 2020iD). The pyrolyzer was attached to a gas chromatograph model 6890N and to a single quadrupole mass selective spectrometer model 5973N (Agilent Technologies Inc., Santa Clara, CA, USA), as described elsewhere [20,21].

We used two different approaches, a direct pyrolysis at 500 °C, or alternatively, firstly, a thermal desorption performed by introducing the sample into a preheated micro-furnace set at sub-pyrolysis temperature of 300 °C. After 2 min the sample was moved to a cold region of the pyrolyzer and the compounds evolved directly injected into the GC-MS for analysis. After a first desorption chromatogram was acquired, the sample was again moved to the pyrolyzer hot area, now pre-heated at pyrolysis temperature of 500 °C, for acquiring a second pyrolysis chromatogram. For both shots, the chromatographic and spectra acquisition conditions were the same.

A low polar-fused silica capillary column (Agilent J&W HP–5 ms Ultra Inert, of 30 m × 250 µm × 0.25 µm film thickness) was used for compound separation and helium, with a controlled flow of 1 mL min$^{-1}$, as carrier gas. The GC oven temperature was set at 50 °C for 1 min and then increased to 100 °C at 30 °C min$^{-1}$, from 100 to 300 °C at 10 °C min$^{-1}$, and then constant at 300 °C for the last 10 min. The mass spectra were acquired at 70 eV ionizing energy. Compound assignments were achieved by single-ion monitoring (SIM) and by comparison with mass spectra libraries (NIST11 and Wiley7).

Thermochemolysis of the melanins with tetramethyl ammonium hydroxide (TMAH) was performed as described by Saiz–Jimenez [22]. The analysis of the thermochemolysis compounds was done using the same instrument previously described and under the same chromatographic and mass spectra acquisition conditions.

## 3. Results

### 3.1. Pyrolysis-Gas Chromatography–Mass Spectrometry of the Melanins from Three Ochroconis Species

The melanins from the three *Ochroconis* species yielded similar pyrograms with hundreds of pyrolysis products when pyrolyzed at 500 °C (Figure 1A–C). A selection of major or representative pyrolysis products are listed in Table 1. The major compounds, common to the three pyrolyzates, corresponded to fatty acids. Peaks were identified as hexadecanoic acid (peak 75) and octadecadienoic, octadecenoic, and octadecanoic acids, (peaks 89–91, respectively).

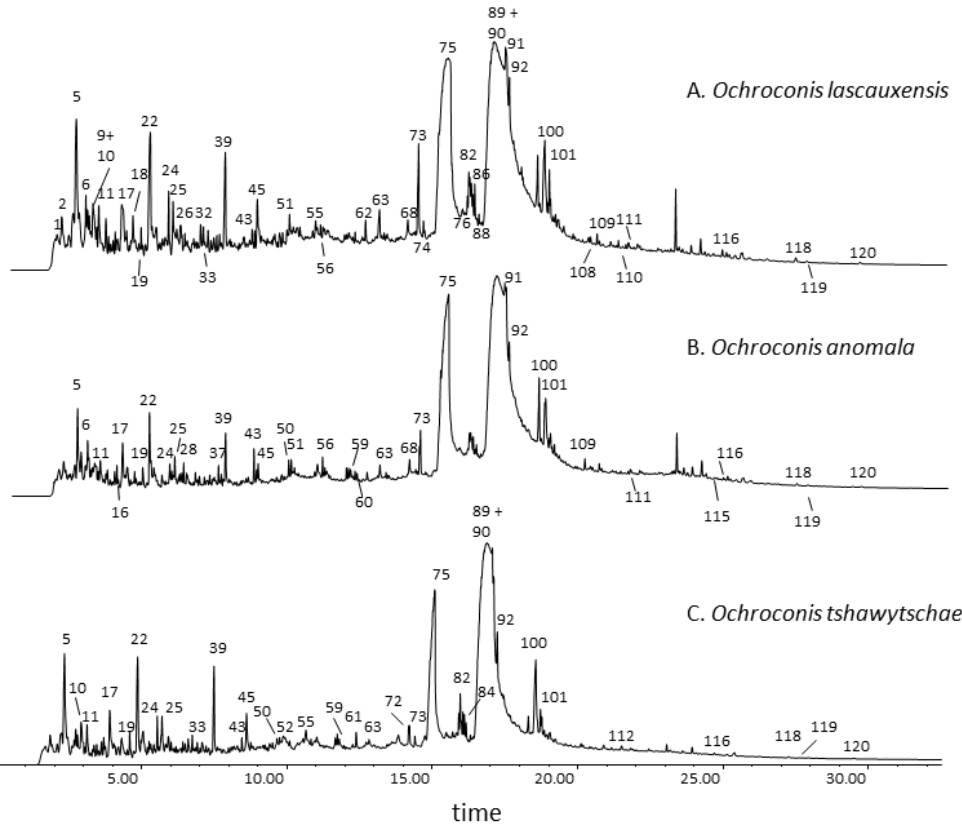

**Figure 1.** Pyrograms of the melanins from three *Ochroconis* species pyrolyzed at 500 °C. Peaks refer to Table 1.

Less abundant fatty acid peaks were assigned to tetradecanoic and pentadecanoic acids (peaks 63, 68), the methyl esters of hexadecanoic, octadecadienoic, octadecenoic, and octadecanoic acids (peaks 74, 84, 85, and 88, respectively), hexadecanoic acid hexadecyl, and octadecyl esters (peaks 117 and 120) in addition to hydroxyoctadecanoic acid γ-lactone (peak 99).

**Table 1.** Pyrolysis products obtained in the pyrolysis at 500 °C of the melanins from three *Ochroconis* species.

| Peak | Compound | Peak | Compound |
|------|----------|------|----------|
| 1 | Methylfuran | 61 | Methylfluorene |
| 2 | Benzene | 62 | Diketodipyrrole |
| 3 | Pyridine | 63 | Tetradecanoic acid |
| 4 | Pyrrole | 64 | *n*-Undecylbenzene |
| 5 | Toluene | 65 | *n*-Octadecene |
| 6 | Furfural | 66 | *n*-Octadecane |
| 7 | Methylpyrrole | 67 | Hexadecanone |
| 8 | Methylpyrrole | 68 | Pentadecanoic acid |
| 9 | Methylpyridine | 69 | *n*-Dodecylbenzene |
| 10 | *n*-Ethylbenzene | 70 | *n*-Nonadecene |
| 11 | Styrene | 71 | *n*-Nonadecane |
| 12 | Dimethylpyridine | 72 | Hexadecanenitrile |
| 13 | Dimethylpyrrole | 73 | Tetradecylfuran |
| 14 | *n*-Propylbenzene | 74 | Hexadecanoic acid methyl ester |
| 15 | Benzaldehyde | 75 | Hexadecanoic acid |
| 16 | Methylfurfural | 76 | Hexahydro-(methylpropyl)-pyrrolo-pyrazinedione |
| 17 | Phenol | 77 | Diketopyrazine (from Pro-Val dipeptide) |
| 18 | Trimethylbenzene | 78 | Diethoxy-tetrahydro-dipyrrolopyrazine |
| 19 | *n*-Butylbenzene | 79 | *n*-Eicosene |
| 20 | Methylpropylbenzene | 80 | *n*-Eicosane |
| 21 | Methylphenol | 81 | Diketopyrazine (from Pro-Val dipeptide) |
| 22 | Methylphenol | 82 | Octadecenenitrile |
| 23 | *n*-Undecane | 83 | *n*-Heneicosene |
| 24 | Benzyl nitrile | 84 | Octadecadienoic acid methyl ester |
| 25 | *n*-Pentylbenzene | 85 | Octadecenoic acid methyl ester |
| 26 | *n*-Ethylphenol | 86 | Octadecanenitrile |
| 27 | Methylbutylbenzene | 87 | *n*-Heneicosane |
| 28 | *n*-Dodecene | 88 | Octadecanoic acid methyl ester |
| 29 | *n*-Dodecane | 89 | Octadecadienoic acid |
| 30 | Naphthalene | 90 | Octadecenoic acid |
| 31 | Dianhydroglucopyranose | 91 | Octadecanoic acid |
| 32 | Vinylphenol | 92 | Hexadecanamide |
| 33 | Benzenepropanenitrile | 93 | *n*-Docosene |
| 34 | *n*-Hexylbenzene | 94 | *n*-Docosane |
| 35 | Methylpentylbenzene | 95 | *n*-Pentadecylbenzene |
| 36 | Picolinamide | 96 | *n*-Tricosene |
| 37 | *n*-Tridecene | 97 | *n*-Tricosane |
| 38 | *n*-Tridecane | 98 | *n*-Hexadecylbenzene |
| 39 | Indole | 99 | Hydroxyoctadecanoic acid γ-lactone |
| 40 | Methylnaphthalene | 100 | Octadecenamide |
| 41 | *n*-Heptylbenzene | 101 | Octadecanamide |
| 42 | Methylhexylbenzene | 102 | *n*-Heptadecylbenzene |
| 43 | *n*-Tetradecene | 103 | Hexahydro-(phenylmethyl)-pyrrolo-pyrazinedione |
| 44 | *n*-Tetradecane | 104 | *N,N*-Dimethyl-octadecenamide |
| 45 | Methylindole | 105 | *N,N*-Dimethyl-octadecanamide |
| 46 | Dimethylnaphthalene | 106 | Docosanenitrile |
| 47 | Phenylhexanone | 107 | Bis-(ethylhexyl)-phthalate |
| 48 | *n*-Octylbenzene | 108 | Oxoheptadecylpyrrolidine |
| 49 | Phenylpyridine | 109 | Tricosanenitrile |
| 50 | *n*-Pentadecene | 110 | Tetracosanenitrile |
| 51 | *n*-Pentadecane | 111 | Oxooctadecadienylpyrrolidine |
| 52 | Dimethylindole | 112 | Oxooctadecenylpyrrolidine |
| 53 | Phenylfuran | 113 | Oxooctadecylpyrrolidine |
| 54 | Naphthalenamine | 114 | Squalene |
| 55 | *n*-Nonylbenzene | 115 | Anthiaergosta-1,5,7,9,22-pentaene |

**Table 1.** *Cont.*

| Peak | Compound | Peak | Compound |
|------|----------|------|----------|
| 56 | *n*-Hexadecene | 116 | Anthiaergostan-5,7,9,16,22-pentene |
| 57 | *n*-Hexadecane | 117 | Hexadecanoic acid hexadecyl ester |
| 58 | Fluorene | 118 | Ergosta-4,6,8(14),22-tetraen-3-one |
| 59 | *n*-Heptadecene | 119 | Stigmasta-3,5-dien-7-one |
| 60 | *n*-Heptadecane | 120 | Hexadecanoic acid octadecyl ester |

The series of *n*-alkanes ($C_6$–$C_{23}$) with *n*-tetradecane and *n*-pentadecane (peaks 44 and 51) as the most prominent compounds, and *n*-alkenes ($C_6$–$C_{23}$) with *n*-tetradecene and *n*-pentadecene (peaks 43 and 50) as the most abundant members were also remarkable.

Aromatic compounds were mainly represented by the series of alkylbenzenes ($C_1$–$C_{23}$) being the most abundant members *n*-butyl-, *n*-pentyl- and *n*-hexylbenzene (peaks 19, 25, and 34). Lesser intensity reached the series of methylalkylbenzenes ($C_1$–$C_{22}$), alkylnaphthalenes ($C_1$–$C_3$), alkylphenanthrenes ($C_1$, $C_2$), and some other compounds such as phenylhexanone (peak 47), fluorene (peak 58), methylfluorene (peak 61), etc. Phenolic compounds were relatively abundant as denoted the peaks of phenol (peak 17), methylphenols (peaks 21 and 22), ethylphenol (peak 26), and vinylphenol (peak 32). A few dialkylphthalates were identified, from which bis-(ethylhexyl)-phthalate (peak 107) was the most significant, likely contaminants introduced during the melanin extraction procedures with reagents and solvents.

Protein pyrolysis products were represented by pyridine, pyrrole and indole derivatives (peaks 3, 4, 7–9, 12, 13, 36, 39, 45, 49, and 52). Characteristic pyrolysis products but with lesser intensity corresponded to a few pyrazine derivatives (peaks 76–78, 81, and 103) and diketodipyrrole (peak 62).

In addition, other nitrogen-containing compounds such as nitriles (peaks 24, 33, 72, 82, 86, 106, 109, and 110 corresponding to benzylnitrile, benzenepropanenitrile, hexadecanenitrile, octadecenenitrile, octadecanenitrile, docosanenitrile, tricosanenitrile, and tetracosanenitrile, respectively), and amides (peaks 92, 100, 101, 104, 105: hexadecanamide, octadecenamide and octadecanamide, *N,N*-dimethyl-octadecenamide, and *N,N*-dimethyl-octadecanamide, respectively) were identified. Some of these nitrile and amide compounds attained relevance, namely those derived from fatty acids with 16 and 18 atoms carbon.

A few compounds were also identified as potential artifacts derived from the reaction of these fatty acids with nitrogen compounds, such as hexadecanoic acid pyrrolidine (peak 108), and oxooctadecadienylpyrrolidine, oxooctadecenylpyrrolidine, and oxooctadecylpyrrolidine (peaks 111–113). Polysaccharide and chitin pyrolysis products were represented by furan derivatives (peaks 1, 6, 16, 25, 53, 73). Dianhydro-*α*-D-glucopyranose (peak 31) and tetradecylfuran (peak 73) were also identified, although this latter could be related with a cyclization of octadecanoic acid.

A number of tetracyclic triterpenes were identified in the pyrolyzate (peaks 115, 117, 118, and 119). These compounds are ergosterol and sitosterol derivatives, common in the fungal cells.

### 3.2. Thermal Desorption-Gas Chromatography–Mass Spectrometry of the Melanin from Ochroconis lascauxensis

Due to the compositional similarity of the three melanins and the abundance of pyrolysis products from different types of materials, we selected the melanin of *Ochroconis lascauxensis* as a model for further analyses. A pyrogram of the bulk melanin revealed an abundance of lipid-, polysaccharide-, and protein-derived compounds as denoted in Figure 1A, which can mask the presence and identification of other structural (aromatic) components. To circumvent this problem, we used a second analytical approach: a double shot pyrolysis involving thermal desorption at 300 °C followed by a pyrolysis at 500 °C. The first thermal step is considered to produce a thermal desorption of lipid compounds, a pyrolysis of polysaccharides and a mild pyrolysis of proteins.

A third analytical approach involved *6N* HCl hydrolysis of the melanins. Acid hydrolysis with constant-boiling at 110 °C for 24 h has been widely accepted as a standard method for removal of proteinaceous materials and carbohydrates [23], which absence could evidence other (aromatic) structural units masked by the high amount of pyrolysis products.

Figure 2A shows the chromatograms of the thermal desorption at 300 °C of the untreated *O. lascauxensis* melanin. Figure 2B depicts the acid hydrolyzed melanin subjected to the same protocol. A list with desorption products at 300 °C of the melanin of *O. lascauxensis* is shown in Table 2.

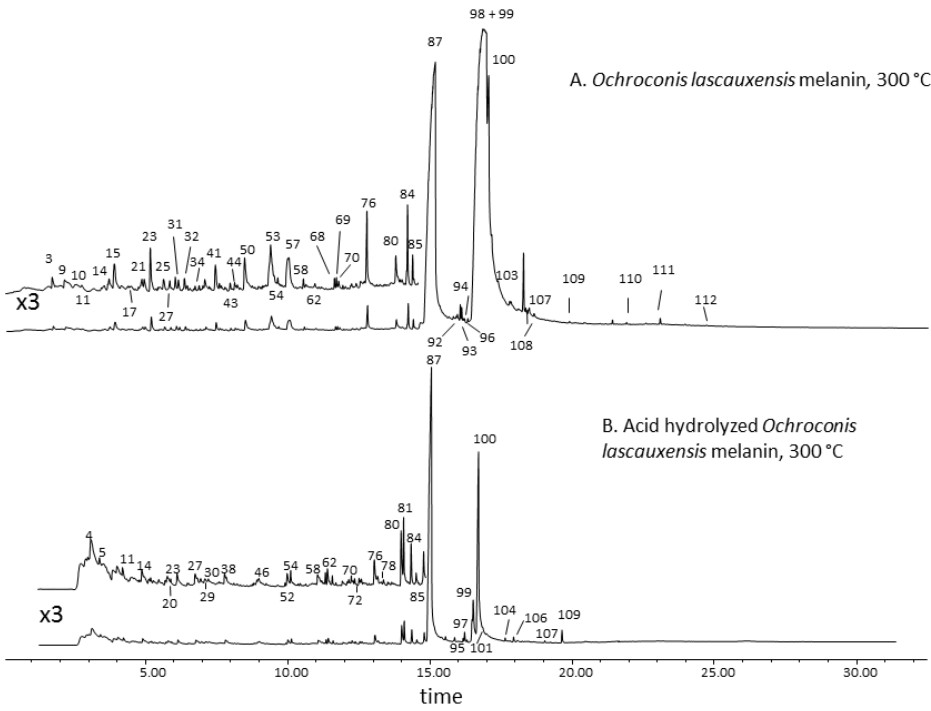

**Figure 2.** Untreated and acid hydrolyzed *Ochroconis lascauxensis* melanin pyrolyzed at 300 °C. Peaks refer to Table 2.

**Table 2.** Pyrolysis products obtained in the pyrolysis at 300 °C of the untreated and acid hydrolyzed *Ochroconis lascauxensis* melanin.

| Peak | Compound | Peak | Compound |
|------|----------|------|----------|
| 1 | Chloromethane | 58 | Dodecanoic acid |
| 2 | Sulfur dioxide | 59 | *n*-Nonylbenzene |
| 3 | Hexanal | 60 | *n*-Hexadecene |
| 4 | Benzene | 61 | *n*-Hexadecane |
| 5 | Dimethyldisulfide | 62 | Diethylphthalate |
| 6 | Toluene | 63 | Methylthiobenzothiazole |
| 7 | Butanoic acid | 64 | Benzophenone |
| 8 | Pyridine | 65 | Phenylhexanoic acid |
| 9 | Furfural | 66 | Tridecanoic acid |
| 10 | Maleic anhydride | 67 | *n*-Decylbenzene |
| 11 | Pentanoic acid | 68 | Heptadecatriene |
| 12 | *n*-Propylbenzene | 69 | Heptadecadiene |
| 13 | Benzaldehyde | 70 | *n*-Heptadecene |
| 14 | Hexanoic acid | 71 | Phenyloctanone |
| 15 | Anhydrodeoxypentenulose | 72 | 2-Pentadecanone |
| 16 | Phenol | 73 | Methyldecylbenzene |
| 17 | *n*-Butylbenzene | 74 | Diketodipyrrole |

**Table 2.** *Cont.*

| Peak | Compound | Peak | Compound |
|------|----------|------|----------|
| 18 | *n*-Butylthiophene | 75 | Phenylheptanoic acid |
| 19 | Acetophenone | 76 | Tetradecanoic acid |
| 20 | Heptanoic acid | 77 | *n*-Undecylbenzene |
| 21 | Methylphenol | 78 | 2-Hexadecanone |
| 22 | Nonanal | 79 | Phenanthrene |
| 23 | Levoglucosenone | 80 | Pentadecanoic acid |
| 24 | Benzeneacetonitrile | 81 | Dibutylphthalate |
| 25 | *n*-Pentylbenzene | 82 | *n*-Dodecylbenzene |
| 26 | *n*-Pentylthiophene | 83 | Hexadecanenitrile |
| 27 | Octanoic acid | 84 | Tetradecylfuran |
| 28 | 2-Decanone | 85 | Hexadecanoic acid methyl ester |
| 29 | Naphthalene | 86 | Hexadecenoic acid |
| 30 | Benzoic acid | 87 | Hexadecanoic acid |
| 31 | Dianhydromannopyranose | 88 | Hexahydro-(methylpropyl)-pyrrolo-pyrazinedione |
| 32 | Dianhydroglucopyranose | 89 | Diketopyrazine (from Pro-Val dipeptide) |
| 33 | Anhydrogalactosan | 90 | Diethoxy-tetrahydro-dipyrrolopyrazine |
| 34 | Anhydromannosan | 91 | Heptadecanoic acid |
| 35 | *n*-Hexylbenzene | 92 | Octadecenenitrile |
| 36 | *n*-Hexylthiophene | 93 | Octadecedienoic acid methyl ester |
| 37 | Phenylbutanone | 94 | Octadecenoic acid methyl ester |
| 38 | Nonanoic acid | 95 | Octadecanoic acid methyl ester |
| 39 | *n*-Tridecene | 96 | Octadecanenitrile |
| 40 | 2-Undecanone | 97 | Hydroxyhexadecanoic acid γ-lactone |
| 41 | Indole | 98 | Octadecadienoic acid |
| 42 | Methylnaphthalene | 99 | Octadecenoic acid |
| 43 | Pentylfuranone | 100 | Octadecanoic acid |
| 44 | Phenylpentanone | 101 | Octadecanoic acid ethyl ester |
| 45 | *n*-Heptylbenzene | 102 | *n*-Pentadecylbenzene |
| 46 | Decanoic acid | 103 | Hexadecanamide |
| 47 | *n*-Tetradecene | 104 | *n*-Tricosene |
| 48 | Biphenyl | 105 | *n*-Hexadecylbenzene |
| 49 | *n*-Tetradecane | 106 | Hydroxyoctadecanoic acid γ-lactone |
| 50 | Galactosan | 107 | Octadecenamide |
| 51 | Phenylhexanone | 108 | Octadecanamide |
| 52 | Octylbenzene | 109 | Bis-(2-ethylhexyl)-phthalate |
| 53 | Mannosan | 110 | Squalene |
| 54 | *n*-Pentadecene | 111 | Hexadecanoic acid dodedecyl ester |
| 55 | *n*-Pentadecane | 112 | Anthiaergosta-1,5,7,9,22-pentaene |
| 56 | Dihydroindolone | 113 | Anthiaergostan-5,7,9,16,22-pentene |
| 57 | Levoglucosan | | |

The untreated melanin is characterized by the presence of huge peaks corresponding to fatty acids: hexadecanoic (peak 87), octadecadienoic (peak 98), octadecenoic (peak 99), and octadecanoic (peak 100) acids. Of minor intensity were the peaks of tetradecanoic acid (76) and the methyl esters of hexadecanoic (85), octadecadienoic (93), octadecenoic (94), and octadecanoic (95) acids. Fatty acid derivatives were the γ-lactone of hydroxyhexadecanoic (peak 97) and δ-lactone of hydroxyoctadecanoic (peak 106) acids and hexadecanoic acid dodedecyl ester (peak 111). Tetradecylfuran (peak 84) was also relevant.

The chromatograms of the melanins pyrolyzed at 500 °C (Figure 1A–C); Table 1 evidenced only a few pyrolysis products from polysaccharides, however at 300 °C (Figure 2A) numerous markers of carbohydrate pyrolysis were clearly observed in the untreated melanin, as denoted the peaks of anhydrodeoxypentenulose (15), levoglucosenone (23), dianhydromannopyranose (31), dianhydroglucopyranose (32), anhydrogalactosan (33), anhydromannosan (34), galactosan (50), mannosan (53), and levoglucosan (57), among others. This change in the pyrolyzate composition was due to the use of a lower temperature, suitable for the pyrolysis of polysaccharides, because 500 °C conducted to an extensive

thermal degradation of the polysaccharide markers, as well as contributed to the excessive production of all types of pyrolysis products, and the difficulty to resolve such a high amount of compounds in the column. Nitrogen derivatives were represented by a few compounds such as indole (peak 41) and octadecanenitrile (peak 95) and minor amounts of diketodipyrrole, pyrazines, and pyrrolopyrazines (peaks 62, 76–78, 81), denoting that the temperature applied was not suitable for a complete pyrolysis of proteins. In addition, peaks of hexadecanamide, octadecenamide, and octadecanamide (101, 104, 105) reached relatively low intensity.

Minor products were represented by the series of *n*-alkanes ($C_9$–$C_{19}$), and *n*-alkenes that reached very low intensity and were mainly evidenced using the ion mass chromatograms. Similar low abundance attained the series of alkylbenzenes ($C_1$–$C_{12}$) and some other compounds such as the peaks of hexadecanamide (101), octadecenamide (104), and octadecanamide (105), a few phenylalkanones (peaks 37, 44, 51), phenol (peak 16), and methyphenol (peak 21). In addition, 2-alkanones ($C_{10}$–$C_{16}$), and a few aldehydes contributed to the identified products.

Figure 2B shows the chromatogram of the acid hydrolyzed melanin after thermal desorption at 300 °C. A list of the products evolved is shown in Table 2. The main differences between the untreated and the acid hydrolyzed melanin resides in the practical absence of carbohydrate and peptide pyrolysis products, the loss of alkylnitriles, alkylamides, and pyrazines, and the decreasing intensity of the fatty acids.

Acid hydrolysis removes carbohydrates and most peptides and proteins, thus preventing the formation of these types of nitrogen-derived compounds. However, acid hydrolysis was not able to remove all fatty acids and the series from butanoic to octadecanoic acids were still present, but only hexadecanoic and octadecanoic acids (peaks 87 and 100) were abundant. The methyl esters of the hexadecanoic and octadecanoic acids (peaks 85, 95), as well as their ethyl esters (peak 101) and hydroxy fatty acids lactones (peaks 97 and 106), were identified as relatively important compounds.

Apart from the fatty acids, there was a minor contribution of aromatic compounds: phenol, benzaldehyde, acetophenone, naphthalene, benzoic acid, biphenyl, benzophenone, phenanthrene, phenyl-fatty acids, and phenyl-ketones.

The chromatogram of the hydrolyzed melanin evidenced as well a minor contribution of the series of alkanes and alkenes ($C_6$–$C_{23}$), alkylbenzenes ($C_1$–$C_{17}$), methylalkylbenzenes ($C_2$–$C_{18}$) (Table 2), and dibutyl- and bis-(ethylhexyl)-phthalates (peaks 81, 109) in Figure 2B. The presence of chlorine and sulfur derivatives (peaks 1, 2, 5, 63), and particularly of a few alkylthiophenes (peaks 18, 26, 36) in Table 2 were noticeable. Chlorine is coming from the hydrochloric acid used for the hydrolysis and sulfur derivatives are related with the percentage of sulfur in the melanin (0.5%) as reported elsewhere [19]. The production of alkylbenzenes and alkylthiophenes in the pyrolysis of fatty acids was already reported by Saiz–Jimenez [24,25].

### 3.3. Second Shot Pyrolysis-Gas Chromatography–Mass Spectrometry of the Melanin from *Ochroconis lascauxensis*

For the second step, performed at 500 °C, the compounds detected were interpreted as fragments produced from the thermal degradation of larger macromolecules or structural units forming the core, once loosely held or volatile compounds have been previously removed at 300 °C.

Figure 3A displays the chromatogram of the melanin pyrolyzed at 500 °C, which previously was subjected to thermal desorption at 300 °C. The pyrolysis products evolved at 500 °C are listed in Table 3.

In the pyrogram of Figure 3A, the fatty acids were drastically reduced, due to the previous thermal desorption, as denoted the low intensity of the major peaks (99: hexadecanoic, 108: octadecenoic, and 109: octadecanoic acids). This was a clear effect of the previous thermal desorption at 300 °C. However, the first 300 °C step had a limited effect on the removal (pyrolysis) of proteins as denoted the identification of pyrrole, pyridine, indole and its alkyl derivatives (e.g., peaks 6, 11, 51, 60, 68), pyrrolopyrazines (peaks 97,

98), diketodipyrrole (peak 81) and the presence of nitriles (peaks 31, 34, 44, 92, 105, 107, 118, 119) and amides (peaks 110, 115, 116). As expected, due to the previous pyrolysis of polysaccharides at 300 °C, carbohydrates markers were almost absent.

Aromatic compounds were represented by the series of alkylbenzenes ($C_1$–$C_{18}$) (e.g., peaks 4, 10, 75, 113, 117), naphthalene and methylnaphthalene (peaks 40, 52), phenanthrene (peak 86), while phenolic compounds were represented by phenol (peak 15), methylphenol (peak 28), methoxyphenol (peak 30), ethylphenol (peak 37), etc.

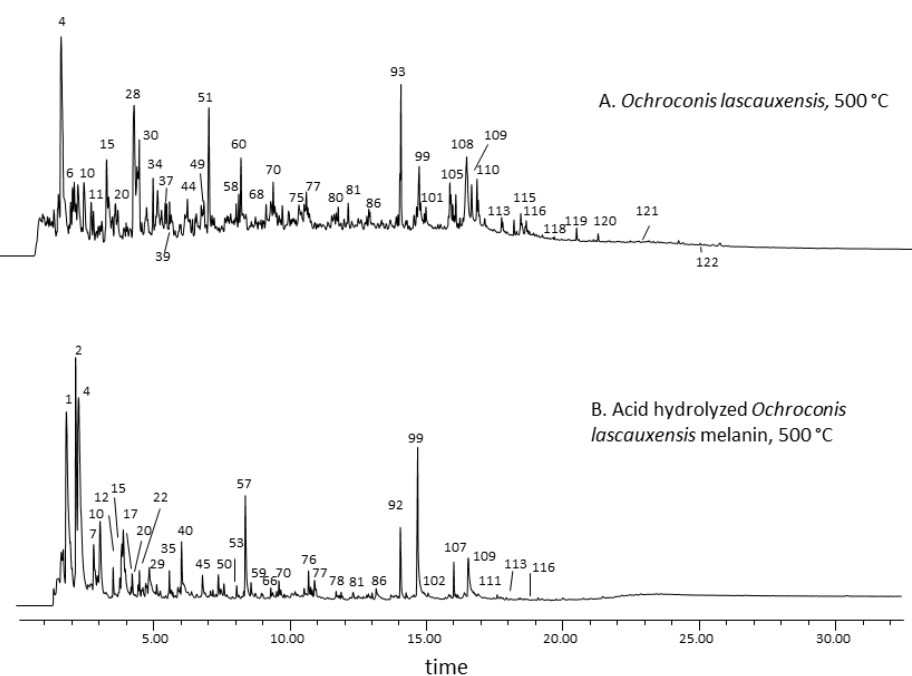

**Figure 3.** Untreated and acid hydrolyzed *Ochroconis lascauxensis* melanin pyrolyzed at 500 °C (second shot). Peaks refer to Table 3.

**Table 3.** Pyrolysis products obtained in the pyrolysis at 500 °C (second shot) of the untreated and acid hydrolyzed *Ochroconis lascauxensis* melanin.

| Peak | Compound | Peak | Compound |
|---|---|---|---|
| 1 | Benzene | 62 | Acenaphthene |
| 2 | Pyridine | 63 | Dimethylnaphthalene |
| 3 | Pyrrole | 64 | Diphenylmethane |
| 4 | Toluene | 65 | Dimethylnaphthalene |
| 5 | Methylpyrrole | 66 | *n*-Octylbenzene |
| 6 | Methylpyrrole | 67 | Methylheptylbenzene |
| 7 | *n*-Ethylbenzene | 68 | Phenylpyridine |
| 8 | *n*-Ethylthiophene | 69 | *n*-Pentadecene |
| 9 | Methylpyridine | 70 | *n*-Pentadecane |
| 10 | Styrene | 71 | Methylbiphenyl |
| 11 | Dimethylpyridine | 72 | Isoindoledione |
| 12 | *n*-Propylbenzene | 73 | Dihydroindolone |
| 13 | *n*-Propylthiophene | 74 | Dibenzofuran |
| 14 | Methylethylbenzene | 75 | *n*-Nonylbenzene |
| 15 | Phenol | 76 | *n*-Hexadecene |
| 16 | Trimethylbenzene | 77 | *n*-Hexadecane |
| 17 | Benzonitrile | 78 | Fluorene |
| 18 | Benzofuran | 79 | *n*-Decylbenzene |
| 19 | Ethymethylbenzene | 80 | *n*-Heptadecane |

**Table 3.** *Cont.*

| Peak | Compound | Peak | Compound |
|---|---|---|---|
| 20 | Indane | 81 | Diketodipyrrole |
| 21 | Indene | 82 | Tetradecanoic acid |
| 22 | *n*-Butylbenzene | 83 | *n*-Undecylbenzene |
| 23 | *n*-Butylthiophene | 84 | *n*-Octadecene |
| 24 | Methylphenol | 85 | *n*-Octadecane |
| 25 | Methylpropylbenzene | 86 | Phenanthrene |
| 26 | Acetophenone | 87 | Phenylbenzofuran |
| 27 | *n*-Undecene | 88 | Pentadecanoic acid |
| 28 | Methylphenol | 89 | *n*-Dodecylbenzene |
| 29 | *n*-Undecane | 90 | *n*-Nonadecene |
| 30 | Methoxyphenol | 91 | *n*-Nonadecane |
| 31 | Benzeneacetonitrile | 92 | Hexadecanenitrile |
| 32 | Methylbenzofuran | 93 | Tetradecylfuran |
| 33 | Methylpropenylbenzene | 94 | Hexadecanoic acid methyl ester |
| 34 | Methylbenzonitrile | 95 | Methylphenanthrene |
| 35 | *n*-Pentylbenzene | 96 | Methylphenanthrene |
| 36 | *n*-Pentylthiophene | 97 | Hexahydro-(methylpropyl)-pyrrolopyrazinedione |
| 37 | Ethylphenol | 98 | Diethoxy-tetrahydro-dipyrrolopyrazine |
| 38 | *n*-Dodecene | 99 | Hexadecanoic acid |
| 39 | *n*-Dodecane | 100 | *n*-Eicosene |
| 40 | Naphthalene | 101 | *n*-Eicosane |
| 41 | Benzoic acid | 102 | Phenylnaphthalene |
| 42 | Vinylphenol | 103 | Octadecadienoic acid |
| 43 | Phenylfuran | 104 | Dimethylphenanthrene |
| 44 | Benzenepropanenitrile | 105 | Octadecenenitrile |
| 45 | *n*-Hexylbenzene | 106 | *n*-Tetradecylbenzene |
| 46 | *n*-Hexylthiophene | 107 | Octadecanenitrile |
| 47 | Methylpentylbenzene | 108 | Octadecenoic acid |
| 48 | *n*-Tridecene | 109 | Octadecanoic acid |
| 49 | *n*-Tridecane | 110 | Hexadecanamide |
| 50 | Methylnaphthalene | 111 | *n*-Tricosene |
| 51 | Indole | 112 | *n*-Tricosane |
| 52 | Methylnaphthalene | 113 | *n*-Hexadecylbenzene |
| 53 | *n*-Heptylbenzene | 114 | Hydroxyoctadecanoic acid γ-lactone |
| 54 | Methylhexylbenzene | 115 | Octadecenamide |
| 55 | *n*-Tetradecene | 116 | Octadecanamide |
| 56 | Methylindole | 117 | *n*-Heptadecylbenzene |
| 57 | Biphenyl | 118 | *n*-Octadecylbenzene |
| 58 | *n*-Tetradecane | 119 | Tricosanenitrile |
| 59 | Methylbiphenyl | 120 | Tetracosanenitrile |
| 60 | Methylindole | 121 | Anthiaergosta-1,5,7,9,22-pentaene |
| 61 | Phenylthiophene | 122 | Anthiaergostan-5,7,9,16,22-pentene |

In addition, other trace compounds, not included in Table 3, such as trimethylphenol, methylmethoxyphenol, methylbenzaldehyde, and ethylbenzenediol, were identified. Other oxygen-containing compounds were acetophenone (peak 26), benzoic acid (peak 41), or phenylfuran (peak 43) (Table 3).

Aliphatic series of compounds, such as alkanes and alkenes ($C_8$–$C_{23}$) with maximum at $C_{14}$ (peak 58) were identified with relative abundance. Ergosterol derivatives (peaks 121, 122) were still present as minor compounds.

Figure 3B shows the chromatogram of the acid hydrolyzed melanin pyrolyzed at 500 °C, after a thermal desorption at 300 °C. The pyrolyzate of this sample was characterized by the intensity of a few products such as benzene (1), pyridine (2), toluene (4), ethylbenzene (7), styrene (10), phenol (15), trimethylbenzene (16), benzonitrile (17), biphenyl (57), hexadecanenitrile (92), hexadecanoic acid (99), octadecanenitrile (107), and octadecanoic acid (109).

Other relevant compounds were the series of alkyl and alkenylbenzenes from $C_2$ to $C_6$ (peaks 7, 12, 22, 35, and 45), although the series enlarged up to $C_{19}$, but they were minor compounds, *n*-undecane (peak 29), *n*-hexadecene (peak 76), and methylnaphthalene (peak 50). It is remarkable the production of minor amounts of sulfur dioxide, dimethyldisulfide, alkylthiophenes from $C_2$–$C_6$ (peaks 8, 13, 23, 36, 46) in Table 3, benzothiophene and phenylthiophene (peak 61), in accord with the sulfur content of the melanin.

Nitrogen derivatives contributed to the pyrolyzate in a lower proportion when compared to the untreated melanin. It was notable the decrease in peaks related to proteins with respect to the untreated (non-hydrolyzed) sample, e.g., indole, methylindole, benzenenitriles, etc. although some nitriles were still present, denoting that the acid hydrolysis did not remove all melanin nitrogen.

The same peak decreasing can be applied to the fatty acids. In fact, both, nitrogen compounds and fatty acids reacted in this pyrolysis protocol to form mainly nitriles and amides derived from the most abundant hexadecanoic and octadecanoic acids, as already observed in the above discussed samples.

In addition, methylphenols (peaks 24, 28), methoxyphenol (peak 30), ethylphenol (peak 37), benzaldehyde, benzophenone, fluorenone, phenanthrenol, etc., were minor pyrolysis products. Other oxygen-containing compounds were exemplified by acetophenone (peak 26), benzoic acid (peak 41), phenylfuran (peak 43), benzofuran (peak 18), methylbenzofuran (peak 32), dibenzofuran (peak 74), and phenylbenzofuran (peak 87) (Table 3).

Aromatic compounds were clearly identified in this sample, namely fluorene (peak 78) and methyl fluorenes, naphthalene and its alkyl derivatives (peaks 40, 50, 52, 63–65), phenanthrenes (peaks 86, 95, 96, 104), biphenyl and methylbiphenyls (peaks 57, 59, 71). Diphenylmethane (peak 64), phenylnaphthalene (peak 102); stilbene, fluoranthene, and terphenyl were minor compounds.

### 3.4. Thermochemolysis with Tetramethyl Ammonium Hydroxide (TMAH) of the Acid Hydrolyzed Melanin from Ochroconis lascauxensis

Figure 4A shows the chromatogram of the acid hydrolyzed melanin subjected to thermochemolysis at 250 °C, and Table 4 lists the compounds identified. The chromatogram was characterized by the abundance of *N,N*-dimethylbenzenamine (peak 18) and other benzenamine derivatives (peaks 11, 17).

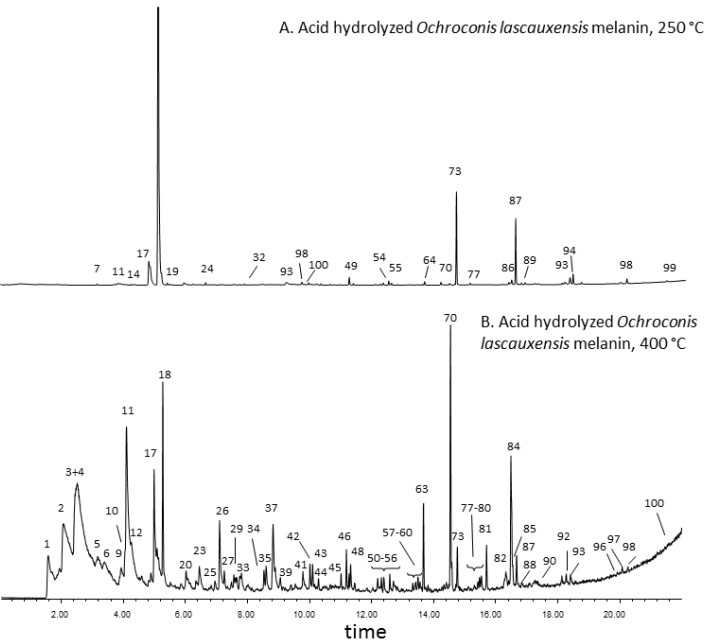

**Figure 4.** Acid hydrolyzed *Ochroconis lascauxensis* melanin pyrolyzed at 250 and 400 °C (double shot) with tetramethyl ammonium hydroxide. Peaks refer to Table 4.

**Table 4.** Pyrolysis products obtained in the pyrolysis at 250 and 400 °C (double shot) with tetramethyl ammonium hydroxide of the acid hydrolyzed *Ochroconis lascauxensis* melanin.

| Peak | Compound | Peak | Compound |
|------|----------|------|----------|
| 1 | Chloromethane + Sulfur dioxide | 51 | *n*-Decylthiophene |
| 2 | Benzene | 52 | *n*-Heptadecene |
| 3 | Pyridine | 53 | *n*-Heptadecane |
| 4 | Toluene | 54 | 2-Pentadecanone |
| 5 | *n*-Ethylbenzene | 55 | Pentadecanal |
| 6 | Styrene | 56 | Tetradecanoic acid methyl ester |
| 7 | Hexanoic acid methyl ester | 57 | *n*-Undecylbenzene |
| 8 | *n*-Propylbenzene | 58 | *n*-Undecylthiophene |
| 9 | Isocyanobenzene | 59 | *n*-Octadecene |
| 10 | Phenol | 60 | *n*-Octadecane |
| 11 | Benzenamine | 61 | 2-Hexadecanone |
| 12 | Trimethylbenzene | 62 | Phenanthrene |
| 13 | Benzofuran | 63 | Hexadecanal |
| 14 | Butanedioic acid dimethyl ester | 64 | Pentadecanoic acid methyl ester |
| 15 | Methylphenol | 65 | Pentyloctylbenzene |
| 16 | *n*-Butylbenzene | 66 | Butylnonylbenzene |
| 17 | *N*-Methylbenzenamine | 67 | Propyldecylbenzene |
| 18 | *N,N*-Dimethylbenzenamine | 68 | *n*-Dodecylbenzene |
| 19 | Octanoic acid methyl ester | 69 | *n*-Nonadecane |
| 20 | *n*-Pentylbenzene | 70 | 2-Heptadecanone |
| 21 | *n*-Dodecene | 71 | Tetramethylfuran |
| 22 | 2-Decanone | 72 | Heptadecanal |
| 23 | Naphthalene | 73 | Hexadecanoic acid methyl ester |
| 24 | Nonanoic acid methyl ester | 74 | Methylphenanthrene |
| 25 | Benzothiazole | 75 | Decylphenol |
| 26 | Quinoline | 76 | Hexadecanoic acid |
| 27 | *n*-Hexylbenzene | 77 | Dibutyl phthalate |
| 28 | *n*-Tridecene | 78 | *n*-Tridecylbenzene |
| 29 | *n*-Tridecane | 79 | 2-Octadecanone |
| 30 | Indole | 80 | *n*-Tridecylthiophene |
| 31 | Methylnaphthalene | 81 | Octadecanal |
| 32 | Decanoic acid methyl ester | 82 | *n*-Tetradecylbenzene |
| 33 | Methylnaphthalene | 83 | *n*-Heneicosane |
| 34 | Heptanedioic acid dimethyl ester | 84 | 2-Nonadecanone |
| 35 | *n*-Heptylbenzene | 85 | Octadecanenitrile |
| 36 | Methylquinoline | 86 | Hydroxyhexadecanoic acid γ-lactone |
| 37 | Biphenyl | 87 | Octadecanoic acid methyl ester |
| 38 | *n*-Tetradecane | 88 | Dodecylphenol |
| 39 | Dodecanal | 89 | Octadecanoic acid |
| 40 | Octanedioic acid dimethyl ester | 90 | *n*-Pentadecylbenzene |
| 41 | *n*-Octylbenzene | 91 | *n*-Hexadecylbenzene |
| 42 | *n*-Pentadecene | 92 | 2-Heneicosenone |
| 43 | *n*-Pentadecane | 93 | Hydroxyoctadecanoic acid γ-lactone |
| 44 | Dodecanoic acid methyl ester | 94 | Eicosanoic acid methyl ester |
| 45 | *n*-Nonylbenzene | 95 | *n*-Heptadecylbenzene |
| 46 | *n*-Hexadecene | 96 | *n*-Octadecylbenzene |
| 47 | *n*-Hexadecane | 97 | Hexadecanephenone |
| 48 | Fluorene | 98 | Bis-(2-ethylhexyl)-phthalate |
| 49 | Diethyl phthalate | 99 | Tetracosanoic acid methyl ester |
| 50 | *n*-Decylbenzene | 100 | Octadecanephenone |

Other compounds, such as hexadecanoic acid methyl ester (peak 73) and octadecanoic acid methyl ester (peak 87) were also significant. In addition, the thermochemolysate was characterized by the presence of minor peaks belonging to the series of fatty acid

methyl esters ranging from $C_3$ to $C_{24}$, and aliphatic dicarboxylic acid dimethyl esters ($C_4$ to $C_9$), as indicated in Table 4. Fatty acid derivatives: hydroxyhexadecanoic acid γ-lactone (peak 86), hydroxyoctadecanoic acid γ-lactone (peak 93) as well as the non-methylated hexadecanoic acid and octadecanoic acids were also identified (peaks 76, 89).

Remarkable is the presence of the series of 2-alkanones from which the most evident compounds were 2-pentadecanone (peak 54), 2-heptadecanone (peak 70), and alkanals from $C_{10}$ to $C_{23}$, maximizing at pentadecanal (peak 55), and a few dialkyl phthalates (peaks 49, 77, and 98).

A second shot of themochemolysis with TMAH at 400 °C of the acid hydrolyzed melanin Figure 4B and Table 4 revealed a complex pyrolyzate dominated by peaks of benzenamine and its methylderivatives (peaks 11, 17, 18), hexadecanal (peak 63), 2-heptadecanone (peak 70), 2-nonadecanone (peak 84), in addition to toluene (peak 4), quinoline (peak 26), biphenyl (peak 37), hexadecane (peak 46), hexadecanoic and octadecanoid acids methyl esters (peaks 73 and 87).

Series of alkylbenzenes ($C_1$–$C_{18}$), methylalkylbenzenes ($C_6$–$C_{11}$), alkylthiophenes ($C_{10}$–$C_{13}$), 2-alkanones ($C_{10}$–$C_{21}$), alkanals ($C_{15}$–$C_{18}$), alkanes ($C_{12}$–$C_{21}$), and alkenes ($C_{12}$–$C_{23}$) were identified as less abundant or minor compounds. A few phenols (peaks 10, 15, 75, and 88), phthalates (peaks 49, 77, and 98), octadecanenitrile (peak 85) and aromatic alkanones (peaks 97 and 100) contributed to the pyrolyzate.

## 4. Discussion

Pyrolysis-gas chromatography-mass spectrometry has been extensively used for investigating the chemical structure (and building blocks) of macromolecular materials such as lignins [26], cutins [27], soil organic matter [22,28–35], etc.

Many studies on the structure of fungal melanins have been published [11,12,17,18,36], however, analytical pyrolysis of melanins has attracted less attention and only a few fungal melanins (e.g., *Aspergillus niger, Epicoccum nigrum, Eurotium echinulatum, Hendersonula toruloidea, Penicillium funiculosum, Stachybotrys atra*) have been studied [37,38].

There are different approaches for analytical pyrolysis (Curie-point pyrolysis in the presence or absence of TMAH, Pyroprobe flash pyrolysis, double-shot pyrolysis–gas chromatography–mass spectrometry, stepwise pyrolysis, etc.). The pros and cons of these techniques have been widely discussed [25,32–35,39–41]. In this work we adopted a double-shot pyrolysis–gas chromatography–mass spectrometry, which includes a first step at 300 °C for thermal desorption and a second step at 500 °C for pyrolysis. In this way it was possible to remove volatile, loosely held products and polysaccharides at 300 °C. The TMAH reagent has been proven to produce some artifacts such as benzenecarboxylic acids [33] and nitrogen-compounds (e.g., triazines) among others [32,42], but conventional pyrolysis has also limitations [24,25,32,43,44].

Melanin is associated to the fungal cell wall. The cell wall is a multilayered structure primarily composed of polysaccharides including β-linked glucan, chitin, mannan and galactofuran. Proteins are also present either covalently linked or more loosely associated [16]. In addition, chitin cross-links to other cell wall polysaccharides and proteins and may form up to 40% of the fungal cell wall, while lipids may reach up to 3% [16,45]. Extraction of fungal melanins with alkaline solvents results in a complex mixture of different materials from the cell walls and the cell itself (fatty acids, sterols, etc.).

In the extraction process with NaOH, different materials present in the fungal mycelia can be associated by covalent bonds or co-extracted and aggregated to the real melanin moiety by hydrogen bonding, van der Waals forces, etc. Thus, the *Ochroconis* melanins have different contributions of polysaccharides, proteins, lipids, and aromatic compounds, whose pyrolysis products are easily identified. These materials have previously been evidenced in a $^{13}$C NMR and $^{15}$N NMR spectroscopic study of these melanins [19].

A review on the type of compounds evolved from the approach used in this work indicates that the pyrolysis products identified in the *Ochroconis* melanins derived from common compounds present in the fungal cells. For instance, homologous series of *n*-

alkanes, fatty acids and fatty acid methyl esters are found in the mycelia of fungi [46,47], as well as polysaccharides [48]. Ergosterol is the major fungal membrane sterol [49]. The different series of compounds found in the pyrolyzates are discussed:

### 4.1. Aliphatic Hydrocarbons

*n*-Alkanes are commonly produced by fungi [46] and the homologous series of *n*-alkanes up to $C_{33}$ were identified [47]. Homologous series of *n*-alkanes, *n*-alkenes and a few *n*-alkadienes have been found in the pyrolyzate of *Ochroconis* melanins. These series of compounds were also identified in the pyrolysis of sodium oleate and sodium linoleate in the presence of sulfur, as well as other series of minor compounds such as alkadienes, alkatrienes, etc. [24]. Decarboxylation converts fatty acids into *n*-alkenes and *n*-alkanes upon direct flash pyrolysis [50]. Pyrolysis of the most abundant fatty acids should produce a series of alkane and alkene doublets below $C_{17}$ due to the ubiquitous presence of $C_{18:1}$ and $C_{18}$ fatty acids in the melanin extract. The identification of up to $C_{24}$ nitrile points out the presence of long chain fatty acids and likely the production of up to $C_{23}$ alkane and alkene doublets. In fact, fatty acids up to $C_{24}$ were identified in the thermochemolysis at 250 °C of the acid hydrolyzed melanin (Table 4). This thermochemolysis also yielded the series of alkane and alkene doublets. Assuming that TMAH protects fatty acids from thermal decarboxylation, these series should be of fungal origin. However, this likely was not the case because the identification of the series of 2-alkanones and alkanals points to a thermal conversion of fatty acids (Table 4). Therefore, to assign a microbial origin to the aliphatic hydrocarbons found in the pyrolyzate might be questionable and likely they can be artifacts occurred during pyrolysis of fatty acids.

### 4.2. Fatty Acids

Fatty acids are synthesized by fungi [46]. Sumner and Evans [51] studied the fatty acids of *Scolecobasidium* spp. The old genus *Scolecobasidium* has been recently revised and most of its species have been transferred to the genus *Ochroconis* [52]. Sumner and Evans [51] reported that different *Scolecobasidium* spp. have the same fatty acid pattern, with hexadecanoic, octadecanoic, octadecenoic, and octadecadienoic acids, as major components and dodecanoic, tetradecanoic, hexadecenoic, and eicosanoic acids as minor components. In fact, these fatty acids are also present in the three *Ochroconis* spp. investigated. Fungi also produce fatty acid methyl and ethyl esters [46,53].

The hexa- and octadecyl esters of hexadecanoic and octadecanoic acids Tables 1 and 2 were attributed to a transesterification process occurring under pyrolysis conditions, but not from thermochemolysis [54].

The origin of some short chain fatty acids and alkanes is doubtful as they could originate from pyrolysis of longer fatty acids. Using Pyroprobe, Faure et al. [55] pyrolyzed undecanoic acid at 620 °C and obtained $C_6$–$C_{10}$ alkenes, $C_8$–$C_{10}$ fatty acids, $C_{10}$–$C_{11}$ alkanals, $C_{11}$ fatty acid methyl ester, and $C_{11}$ γ-lactone. Pyrolysis at 650 °C of oleic acid in the same pyrolytic unit yielded fatty acids with shorter C chain ($C_7$–$C_{16}$) and alkenes, aldehydes, alcohols, and ethers were also identified [56]. In addition, the lactones identified in the pyrolyzates of the melanins must have their origin in the major fatty acids $C_{16}$ and $C_{18}$ [57]. These lactones were even identified in the thermochemolysis with TMAH, either at 250 or 400 °C. This methylation procedure yielded non-methylated fatty acids and phenols (Table 4).

### 4.3. Other Aliphatic Compounds

*n*-Alkan-2-ones in the range $C_{10}$–$C_{19}$ were observed in the melanins, mainly in the first thermal desorption shot (300 °C) and in the thermochemolysis at 250 and 400 °C. Methyl ketones can be produced by fungi by decarboxylation of β-keto acids [58], however, these ketones probably corresponded to thermal reaction products of fatty acids [59]. This was also supported by the occurrence of *n*-alkan-2-ones with a similar distribution that the lipids in the first shot of the non-hydrolyzable organic fraction of a soil [35].

Almendros et al. [60] reported that, pyrolysis at high temperature increased the yield of aromatic compounds and furans, whereas the release of ketones was more characteristic of samples heated at lower temperatures.

Heating (300 °C) of benzoic acid and fatty acids in the presence of iron salts yielded alkyl phenyl ketones, benzophenone, and benzaldehyde [60]. Thus, probably, the identification of alkyl phenyl ketones in the melanins can be explained by reaction of fatty acids in the presence of aromatic rings. The $C_{16}$ and $C_{18}$ alkyl phenyl ketones were evident in the second shot thermochemolysis at 400 °C of the acid hydrolyzed melanin.

The presence of 2-alkanones among the pyrolysis products points to their formation by a mechanism involving decarboxylation and dehydration [59]. The hypothesis that *n*-alkanones occur by thermal cleavage of ether bonds in the pyrolysis of kerogens and algaenans (biopolymers of microalga cell walls) [41,61] likely cannot explain their presence in the pyrolyzate of the extracted melanins due to the fact that these highly aliphatic biopolymers has not been reported in fungi. However, the presence of other type of ether bonds in the melanins cannot be discarded.

Alkylamides are pyrolysis products formed by reaction between the ammonia evolved from the pyrolysis of proteins and the fatty acids. Alkylnitriles are formed via dehydration of alkylamides [62,63]. Nitriles and amides have been identified in different pyrolyzates of the melanins and their presence indicated that acid hydrolysis did not remove all melanin nitrogen and the same can be applied to the fatty acids. In this way, the most abundant fatty acids (hexadecanoic and octadecanoic acids) yield the corresponding alkylamides and alkylnitriles. In addition, alkylamides, alkylnitriles, and fatty acids were abundantly recovered in the pyrolyzates of the microalga *Aurantiochytrium* sp., whose major components were carbohydrates, proteins, and lipids [64].

Considering the data discussed, everything points to the fact that likely many of the alkanes, alkenes, ketones, alkanals, γ-lactones, amides, and nitriles could be secondary pyrolysis products from the abundant fatty acids present in the extracted melanins.

### 4.4. Proteins

Pyrolysis of proteins, peptides and amino acids has been widely investigated. The formation of diketopiperazines (DKP) and diketodipyrrole is a major pyrolytic pathway for proteinaceous materials [65,66]. Smith et al. [67] reported that pyrolysis of dipeptides readily expelled water to form DKPs by cyclization of the dipeptide. Torri et al. [68] described the formation of different protein pyrolysis products from microalgae, including in addition to DKPs, indole-, pyrrole-, imidazole-derivatives, as well as alkylnitriles and alkylamides. Indoles, pyrroles, and phenols are pyrolysis products that can be produced from the amino acids tryptophan, phenylalanine, or tyrosine. The pyrolysis of collagen (unpublished data) gave as major pyrolysis products diketodipyrrole and the DKPs identified in the melanin of *Ochroconis*. Although it seems clear that the DKPs identified are produced from the pyrolysis of proteins, it should be noted that a few DKPs can also be synthesized by fungi [69,70].

Chitin is a constituent of the cell walls that is usually associated to proteins when extracted from fungal mycelia. The protein content of some fungal chitin extracts ranged between 10 and 15% [71]. Pyrolysis of chitin and cross-linked proteins from invertebrate cuticles produced diketodipyrrole and DKPs [65], as also observed in the melanins of *Ochroconis* spp. These melanins contain nitrogen refractory to acid hydrolysis as revealed the identification of nitrogen derivatives, namely quinolines, diketodipyrrole, and octadecanenitrile, even in the second shot of the thermochemolysis.

### 4.5. Carbohydrates and Polysaccharides

The pyrolysis of carbohydrates and polysaccharides has been extensively studied and their pyrolysis products are well-known [72,73]. The initial thermal decomposition of cellulose is the depolymerization of the polymer to form various anhydrosugar derivatives, among which levoglucosan is the most prevalent [74].

Pyrolysis of soil polysaccharides produced low molecular weight furan compounds, levoglucosenone and anhydrosugars [28] but not long chain alkylfurans. The tetradecylfuran found in the pyrolyzate of the melanins could result from the intramolecular cyclization of a $C_{18}$ fatty acid. This compound has also been identified in humic acids, a soil macromolecule with high percentages of fatty acids [75,76]. Furan derivatives such as phenylfuran, benzofuran, methylbenzofuran, dibenzofuran, and phenylbenzofuran were identified in the pyrolyzate (500 °C) of the *O. lascauxensis* acid hydrolyzed melanin. Some authors reported that these compounds are pyrolysis products from polysaccharides [77,78], while for others considered that alkylbenzofurans are lignin pyrolysis products [79]. Their detection after polysaccharides removal in the first shot at 300 °C would indicate the occurrence of aromatization processes either during the first and second shot from other fungal compounds still present in the melanins.

*4.6. Sterols*

Ergosterol is the major fungal membrane sterol [49], although many other sterols (up to 29) were identified in melanized fungi by Méjanelle et al. [80]. The sterols identified from the *Ochroconis* melanins are derivatives from ergosterol and sitosterol. However, the mass spectrometry system operating in selected ion monitoring mode (m/z 337, 363, and 396) was unable for detecting ergosterol in the *Ochroconis* melanins. The sterols identified in the melanin, pyrolyzate (Table 1) were either produced by the fungus, since oxidation of dehydroergosterol gives ergostatetraenone [81], or alternatively they were products of thermal reactions. For instance, stigmastadienone can be found among the thermal desorption products (280 °C) of a biofilm [82] or in the dichloromethane-methanol extracts of soil organic matter after a wildfire [83]. Some other unidentified sterols were observed in the pyrolyzate of the three *Ochroconis* spp. and could results from thermal reaction of original sterols. No sterol derivatives were identified in the acid hydrolyzed melanin, suggesting that the sterols were entrapped in or adsorbed to the melanin components, but released upon acid hydrolysis.

*4.7. Alkylbenzenes, Alkylthiophenes, and Alkylnaphthalenes*

Some fungal melanins produced upon pyrolysis *n*-alkylbenzenes and alkylnaphthalenes [37]. The homologous series of *n*-alkylbenzenes were also found in pyrolyzates of geopolymers [84]. Saiz–Jimenez [24,25,43] pointed out that the formation of the homologous series of *n*-alkylbenzenes were artifacts generated during pyrolysis from unsaturated fatty acids. In fact, pyrolysis of free fatty acids, sodium salts of fatty acids, and triglycerides, in the presence of elemental sulfur, were largely transformed into homologous series of alkylbenzenes and alkylthiophenes [24]. The fungal melanins investigated have 0.5% of sulfur. It has been proved that under the analytical conditions used to study humic substances (soil macromolecule with similar components that some fungal melanins, e.g., fatty acids and sulfur, among others), thermal cyclization and aromatization of fatty acids take place in the presence of either elemental or organic sulfur. Identification of alkylbenzenes (as major compounds), alkylthiophenes and alkylnaphthalenes in pyrolyzates of soil humic substances could be related to thermal reactions of aliphatic precursors [25].

Interestingly, alkylbenzenes were found in the first shot of the melanin pyrolyzate but not alkylthiophenes. However, in the second shot alkylthiophenes were identified. It can be speculated that the temperature of the first shot was not high enough to produce thiophenes. Saiz–Jimenez [25] obtained alkylthiophenes from fatty acids pyrolyzed at 500 °C in the presence of sulfur. In the melanins, alkylthiophenes were identified in the second themochemolysis shot at 400 °C but not in the first one at 250 °C, which confirms its dependence from the temperature used and the absence of TMAH protection regarding the production of these alkyl aromatic secondary pyrolysis products.

### 4.8. Oxygen-Containing Compounds

Besides phenol, the pyrolyzates showed many other phenolic compounds. A few phenols were considered as pyrolysis products of proteins such as phenol and methylphenol [85] but they are also major products in the pyrolyzates of phenolic polymers. In fact, phenolic compounds were obtained in high amounts in the pyrolysis of the melanin of *E. echinulatum*, which is formed extracellularly by oxidation of phenols [18,78] as shown in the Supplementary Figure S1. Strong signals in the aromatic region (25% of the total intensity) were evidenced in the $^{13}$C NMR spectrum of this melanin while about 16% was noticed in the *O. lascauxensis* melanin [19]. This suggests that aromatic compounds could be involved in the formation of the *Ochroconis* melanin core, as discussed previously for other fungi [18].

Other oxygen-containing compounds (acetophenone, benzophenone, benzofurans, dibenzofuran, etc.) have different origins. For some authors, acetophenone is a pyrolysis product from lignin and benzofurans are pyrolysis products from polysaccharides or lignins [79,86], while benzophenone has been recorded in the pyrolyzates of soil organic matter [35]. Obviously some of these pyrolysis products should have a different origin, pointing to rearrangement of carbohydrates or fatty acids.

The identification of alkylphenols, such as decylphenol, dodecylphenol, and phthalates in the thermochemolysis (Table 4) could likely be related to additives present in the plastic wares used in the protocol extraction of melanins and TMAH products.

### 4.9. Acid Hydrolysis

Acid hydrolysis is a technique used to purify fungal melanins [15,36]. It has been reported that this procedure removes polysaccharides, proteins, lipids and loosely held compounds from complex bio- and geomacromolecules [23,38]. However, this procedure did not remove all protein nitrogen and fatty acids, as denoted by the presence of fatty acids and nitriles in the chromatograms of the acid hydrolyzed melanin pyrolyzed at 500 °C (Figure 3B and Table 3), and when protected with TMAH (Figure 4B and Table 4). Remarkable is the absence of nitriles in the acid hydrolyzed melanin pyrolyzed at 300 °C, and in the thermochemolysis at 250 °C which suggests that in spite of having ammonia from N terminal amino acids, these temperatures are not high enough for causing the reaction leading to nitriles that occurs when pyrolyzed at 400–500 °C. De la Rosa et al. [87] reported that alkylnitriles were not observed when soil organic matter was pyrolyzed at 340 °C, but they appeared in a second shot at 520 °C.

Zang and Hatcher [88] treated with 6*N* HCl lake sediments to get a sample enriched in refractory nitrogen. The chromatogram revealed the presence of DKP derivatives among the pyrolysis products and the amide signal was still present in the $^{15}$N NMR spectrum, which denoted that the sample still contained organic nitrogen. A similar result was obtained by Knicker et al. [89] in the hydrolysis residue of an algal sapropel subjected to thermochemolysis with TMAH. The authors suggested that the amide-N in the $^{15}$N NMR spectrum originated from peptide-like structures that have survived diagenetic alteration as well as alkali and acid hydrolyses.

### 4.10. Conventional Pyrolysis vs. Thermochemolysis

When compared conventional pyrolysis (Figures 1–3 and Tables 1–3) and thermochemolysis (Figure 4 and Table 4), it becomes evident that a part of methylated compounds present in the thermochemolysate corresponded with their counterpart in conventional pyrolysis, namely fatty acids. The second group represented new compounds that were not detected without TMAH (e.g., aliphatic dicarboxylic acids) and a third group comprised compounds that were present in both thermal treatments (e.g., alkane and alkene doublets, alkylbenzenes, alkylthiophenes, etc.). The pyrolysis products comprised in the first and third groups have been widely discussed.

In the second group are included short chain aliphatic dicarboxylic acid dimethyl esters. Interestingly, butenedioic and butanedioic dimethyl esters were the main ther-

mochemolysis products from polymaleic acid, a polycarboxylic aliphatic polymer prepared by a pyridine-catalyzed homopolymerization of maleic anhydride [28,32]. The polymer contains units of four carbon atoms in a chain highly substituted with carboxylic groups [90]. Conventional pyrolysis (500 °C) of polymaleic acid yielded different furandiones, maleic anhydride, phthalic anhydride, and carboxybenzeneacetic acid indicating that pyrolysis of aliphatic polycarboxylic acids produced cyclization of the aliphatic polymer [32], which is avoided by the methylation. The diacid dimethyl esters probably reflect alkyl chains bonded to a macromolecular network by one or two ester linkages.

Thermochemolysis of the melanins was characterized by the abundant presence of *N,N*-dimethylbenzenamine and other benzenamine derivatives. Joll et al. [91] found that some of the main products from thermochemolysis of 2-aminobenzoic acid were benzenamine, *N*-methylbenzenamine, and *N,N*-dimethylbenzenamine, denoting that decarboxylation occurred in addition to methylation of the amino group. This fact suggests that the abundant benzenamine derivatives found in the thermochemolysate of the acid hydrolyzed melanin could derive from aminobenzoic units. However, this assertion can be disregarded because in the thermochemolysis of the aliphatic polyesters cutin and suberin we identified up to $C_{30}$ fatty acids and aliphatic hydrocarbons, in addition to benzenamine derivatives (data not shown), in spite that no nitrogen compounds are present in these polyesters. These data indicate that benzenamine derivatives in the melanins are artifacts produced by rearrangement of the TMAH reagent.

Surprisingly, most of the thermochemolysis papers neglected the artifacts introduced with the TMAH and only a few references can be found [33,92]. Many artifacts contain up to four N atoms [93], and Frazier et al. [42] included triazines, pyrimidinediones, pyrolidinediones, and caffeine among the artifacts produced in pyrolysis from dissolved organic matter in freshwaters.

### 4.11. The Chemical Structure of Ochroconis Melanins

If we assume that the fungal melanins are a mixture of compounds and polymers extracted from the fungal cells and that most of the compounds released by thermal desorption or pyrolysis are not part of the real melanin moiety, what is the chemical structure of this fungal melanin?

The decades from the 1960s to the 1980s of the last century were characterized by the efforts to investigate the reaction between phenols and amino acids and proteins, considering that this was the formation process of soil humic substances and fungal melanins [11,17,18,94–96]. Briefly, phenols are readily oxidized to quinones. Phenols and quinones interact with proteins reversibly by hydrogen bonding, or irreversibly by covalent bonding. Phenolic compounds and amines are linked together by oxidative coupling. Products of oxidative coupling reactions result from C-C and C-O coupling of phenols and from C-N and N-N coupling of amines [95,96] although C-S from thiols were also considered [17]. This originated the formation of black polymers from which part of the amino acids can be recovered after acid hydrolysis. The percentage of residual nitrogen after acid hydrolysis in some fungal melanins ranges between 5 and 56% and that of sulfur from 41 to 75% [12,17,18]. This denotes that the coupling C-N and C-S is resistant to acid hydrolysis. According to Haider et al. [94] the reaction products of phenols with amino acids were stable against acid hydrolysis and it was found that all amino acids, except the N-terminal bound to oxidized phenols, could be hydrolyzed. It is highly probable that this type of units, formed by the reaction of phenols and quinones with proteins are present in the melanin.

The hypothesis of a phenolic-protein polymer for some fungal melanins is supported by the data obtained from the pyrolysis of *E. echinulatum* melanin (Supplementary Figure S1), and their $^{13}$C and $^{15}$N NMR spectra [19]. In addition, the pyrolysis-mass spectra of this melanin and a synthetic phenolic-protein polymer were almost identical [97]. Although the *Ochroconis* melanins showed a relatively high aromatic character (16 % of the total $^{13}$C intensity) and carboxyl and carbonyl functional groups (16%), *E. echinulatum* reached the

highest aromatic character (25%) and carboxyl and carbonyl functional groups (18%) of all the melanins investigated [98]. Therefore, the *E. echinulatum* melanin is representative of a phenolic-protein polymer. In fact, the major pyrolysis products were phenol and benzenediol derivatives including a few anthraquinones previously identified in the culture medium [18] and linked to the polymer, as well as some pyrolysis products from proteins, polysaccharides, and lipids.

Another model melanin was extracted from the fungus *Cryptococcus neoformans*. This DOPA-melanin, upon pyrolysis, should yield indole-derivatives and phenols. Pyrolysis showed an abundance of fatty acid derivatives and sterols, natural component of the cells, co-extracted with the melanin (Supplementary Figure S2). Fatty acids were represented by unmethylated fatty acids and their methyl and propyl esters, as well as fatty acids amides. The covalent binding of *C. neoformans* melanin to aliphatic moieties (polysaccharides, chitin and chitosan, and lipids) has been shown by NMR analyses [99]. In addition, melanin-associated proteins were identified from *C. neoformans* melanin granules [99]. The presence of alkanes, alkenes and alkylbenzenes suggest a breakdown or cyclization from fatty acids upon pyrolysis, as reported by Saiz–Jimenez [24,25]. Phenols and nitrogen derivatives were also relatively abundant, as well as polycyclic aromatic hydrocarbons, which might be evidence of a DOPA core.

Previous studies on solid-state cross polarization magic-angle spinning $^{13}$C and $^{15}$N NMR and SERS spectroscopy of the melanins extracted from the mycelia of *O. lascauxensis*, *O. anomala*, and *O. tshawytschae* revealed that they have similar spectra [19]. Solid-state $^{15}$N NMR spectra of the melanins showed an intense signal in the peptide and amide region revealing that a major part of their organic N is bound in peptide structures, and the presence of aromatic N heterocyclic signals is negligible, if any. In addition, there were no bands assigned to pyrrole and indole ring vibrations in the SERS spectra. These data were also supported by the tests of melanin inhibition. Neither kojic acid nor tricyclazole inhibit the synthesis of *Ochroconis* melanin, which indicates that it is not a DOPA or DHN melanin. Therefore, according to De la Rosa et al. [19] the chemical structures of these melanins are not related to DOPA, 5,6-dihydroxyindole or DHN precursors. The analytical pyrolysis of *Ochroconis* melanins also supports this assumption because indole or naphthalene derivatives were not the major pyrolysis products.

Natural DOPA melanins (e.g., from *Sepia officinalis* and *C. neoformans*) and synthetic eumelanins share a common indole-based aromatic core, as demonstrated using $^{13}$C and $^{15}$N NMR [100,101]. The same NMR approach revealed that an indole-based aromatic core could not be found in *Ochroconis* melanins [19], but a high peptide content. Relative high peptide contents (>70% of the total intensity in the $^{15}$N NMR spectra) were also found in other fungal melanins (*Aspergillus glaucus, Hendersonula toruloidea, Ulocladium atrum, Trichoderma harzianum, E. echinulatum*, etc.). This indicates that most, if not all, of the organic nitrogen in these fungal melanins were bound in peptide-like structures [98].

The pyrolysis of DOPA melanins yield as major products those derived from an indole core: pyrroles, pyridines, toluene, phenols, and indoles [102,103]. Although the pyrolysis of a few fungal melanins (e.g., *Aspergillus niger, Aspergillus sydowii, Penicillium funiculosum, Coprinus micaceus, Stachybotrys chartarum*) produces the same compounds [37], they are found in comparatively lower amounts, which easily can be ascribed to pyrolysis of proteins, co-extracted or linked to the melanin [37,94,104]. This is also the case of the *Ochroconis* melanins, as shown by NMR [19].

What other possibilities exist for the chemical structure of *Ochroconis* melanins? Some authors reported fungal melanins based on DHN (e.g., *Aspergillus fumigatus, Neurospora crassa*) [105,106]. A structure based on the polymerization of 1,8-dihydroxynaphthalene should yield upon pyrolysis major products derived from the naphthalene core. No conclusive pyrolysis data were found on this type of melanin. However, Beltrán–García et al. [107] carried out a structural analysis of the melanin of *Mycosphaerella fijiensis* using linear-MALDI-TOF mass spectrometry revealing a molecular mass of around 8000 Da,

corresponding to 50 units of 1,8-DHN, and its spectrum showed a well-defined spacing of 161.8 Da between peaks, associated with 1,8-DHN units.

To the light of the pyrolysis data of the acid hydrolyzed melanin, a core of 1,8-DHN units for *Ochroconis* melanins should be disregarded, because the intensity of peaks related with naphthalene derivatives are similar to those obtained in the pyrolysis of proteins.

It can be speculated that the structural units of *Ochroconis* melanins would be based on the polymerization of phenols and quinones, as suggests the identification of a few pyrolysis products (phenol, methylphenols, ethylphenol, methoxyphenol, benzaldehyde, benzophenone, fluorenone, phenanthrenol) found in the acid hydrolyzed melanin (Figure 3B and Table 3). This could be indicative of a possible pyomelanin formation. In this melanin type, the process starts when homogentisic acid is oxidized to benzoquinone acetate and polymerizes to form pyomelanin. This is similar to the pathway described for the fungal melanins of *Aspergillus sidowii* [17] and *Hendersoluna toruloidea* [36], in which 2,5-dihydroxybenzoic acid (gentisic acid), among other phenols, is oxidized by a phenoloxidase to benzoquinone with the formation of melanin. However, phenols and quinones reached low intensity in the pyrogram of the *Ochroconis* melanin, which could be explained by the relative high amount of associated compounds (polysaccharides, proteins, and lipids). Anyway, further studies are needed to prove if *Ochroconis* pigments are pyomelanins synthesized from tyrosine through *p*-hydroxyphenylpyruvate and homogentisic acid [105]. Vasanthakumar et al. [108] studied the synthesis of pyomelanin by *Penicillium chrysogenum* and observed that inclusion of DOPA melanin inhibitor (kojic acid) and DHN-melanin inhibitor (tricyclazole) in the tyrosine medium did not inhibit the pigment formation, in a way similar to that found in *Ochroconis* [19]. Interestingly, *C. neoformans* can synthesize melanin from homogentisic acid, but there were significant differences between the melanin produced from homogentisic acid and the classic black pigments produced from DOPA and other catecholamines, as deduced from the presence of different compounds after permanganate oxidation [109].

## 5. Current Challenges and Future Perspectives

The critical problem in the study of melanins is the obtaining of a pure macromolecule without mixture of any other cell components. Common isolation procedures imply an extraction with an alkaline solvent and further precipitation in acid media [13,19,110]. This protocol co-extracts a wide diversity of components, associated with the fungal cell wall. A different multi-step extraction procedure has been used by other authors seeking for the method of obtaining pure melanin. This procedure, applied to the extraction of *C. neoformans* melanin, includes an enzymatic digestion of cell-wall components, followed by protein denaturation and subsequent degradation, chloroform extraction to remove lipids and acid hydrolysis [15,101]. This melanin still contained some associated cell wall components, see Supplementary Figure S2. Therefore, a challenge is to get a suitable extraction procedure for the fungal melanins.

A step further would be the use of synthetic melanin and the study of their pyrolysis products. Attempts to obtaining useful DHN melanin models (free DHN polymer or DHN-peptide polymer) failed because the pyrograms revealed a lot of impurities likely introduced during the chemical synthesis (protein from the enzymes used in the polymerization, alkanes, fatty acids, etc.) and the study of the pyrolysis products was inconclusive.

Alternatively, chemical degradation of the *Ochroconis* melanin versus other DOPA and DHN melanins seems to be needed in order to identify specific markers of each melanin-type and disclose the structural problem. However, as reported by Pralea et al. [103], "due to the high heterogeneity of melanins, their analytical characterization can be a challenging task ... Our thorough comparative evaluation of analytical data published so far on melanin analysis has proven to be a difficult task in terms of finding equivalent results, even when the same matrix was used".

Another approach is the study of *O. lascauxensis* genome in search of genes involved in the synthesis of melanin or pyomelanin. Unfortunately the literature only reported

the in silico genome analysis of *Ochroconis mirabilis* [111], an opportunistic pathogen. This analysis revealed that the fungus has the potential ability to synthesize melanin via the 1,8-dihydroxynaphthalene (DHN)-melanin pathway and to produce trichothecene (mycotoxin). The possible synthesis of trichothecene is of interest because it has been suggested that the melanin of *Stachybotrys chartarum* can be based on the polymerization of trichothecenes, as evidenced pyrolysis and $^{13}$C NMR data [19,37]. In fact, *S. chartarum* synthesizes, in addition to phenols, a high number of mycotoxins, among which there are the sesquiterpenoid trichothecenes and the diterpenoid atranones [11,112].

The $^{13}$C NMR data of *S. chartarum* melanin agree with the possible involvement of mycotoxins (trichothecenes and atranones, among others), which possess a terpenoid structure having methylene chains and carbonyl groups. It can be speculated that the reactive groups from mycotoxins and phenols can undergo polymerization reactions resulting in a complex macromolecule with a marked aliphatic character [19,113]. In addition, the pyrolysis of this melanin produced compounds such as series of alkylcyclohexenes, alkylbenzenes, and alkylnaphthalenes. These compounds are secondary thermal reaction products formed upon pyrolysis of aliphatic compounds through cyclization and aromatization reactions [24,37].

According to these data, another possible pathway for melanin formation is the polymerization of phenols and terpenoids. Recently, the formation of a pheomelanin-like polymer has been reported in the fungus *Plenodomus biglobosus* [114] which considerably complicates the situation, and leaves the chemical structure of fungal melanins even more open.

## 6. Conclusions

Analytical pyrolysis is a tool widely used for investigating the chemical structure of macromolecules. This technique has been applied with success to natural and synthetic polymers. However, for complex macromolecules, such as fungal melanins the pyrolysis presents some drawbacks. In fact, they are macromolecules that rarely can be obtained in a pure grade and contains a mixture of polysaccharides, chitin, proteins, nucleic acids, and lipids, among other materials. Analytical pyrolysis permitted the characterization of compounds co-extracted or linked to the melanin, however, the high number of evolved pyrolysis products masks the analysis of the melanin building blocks due to complex pyrograms with hundreds of compounds. This makes it difficult to determine the type and structure of the melanin. In addition, the data revealed that under the conditions usually employed in the pyrolysis of melanins, the technique presented serious limitations due to the diversity of components that suffer extensive thermal degradation through secondary reactions with the production of a considerable number of artifacts. This has been proved in the pyrolysis of *Ochroconis* melanin.

Different pyrolysis approaches and derivatization techniques were not able to provide information on the melanin structural units. Even when the melanin was purified by acid hydrolysis to remove most of the fungal cell components, the melanin was still elusive to provide insight on their main building blocks. The comparison of pyrograms and pyrolysis products from *Ochroconis* melanin with those from other well-characterized fungal melanins showed clear differences and it was hard to conclude if the studied melanin is based on a phenolic-protein polymer (pyomelanin), DOPA or DHN melanin. Further studies focusing on chemical degradation of the macromolecules and the fungal genome and the presence of specific genes involved in melanin synthesis might provide some more insights.

To conclude, the used pyrolysis and thermochemolysis approaches give rise to considerable uncertainty, as a structural model cannot be proposed on the basis of pyrolysis products suspected to be secondary thermal reaction products. Anyway, the statement issued 25 years ago [32] is still valid: "it is of the utmost importance to bear in mind the real value of the pyrolysis products and to what extent they are representative of or can be related to structural units and one has to be cautious in the transfer of data from pyrolysis to the macromolecule when the establishment of a chemical structure is intended".

**Supplementary Materials:** The following are available online at https://www.mdpi.com/2076-341 7/11/3/1198/s1, Figure S1: Py-GC-MS chromatogram of *Eurotium echinulatum* melanin, Figure S2: Py-GC-MS chromatogram of *Cryptococcus neoformans* melanin.

**Author Contributions:** Conceptualization, C.S.-J.; methodology and data curation, P.M.M.-S.; J.A.G.-P.; B.H.; writing—original draft preparation, C.S.-J.; writing—review and editing, C.S.-J.; All authors have read and agreed to the published version of the manuscript.

**Funding:** This research received no external funding.

**Data Availability Statement:** Data supporting reported results can be obtained from the authors.

**Acknowledgments:** *Cryptococcus neoformans* melanin was kindly supplied by Radames J. B. Cordero, Johns Hopkins University, Baltimore, USA.

**Conflicts of Interest:** The authors declare no conflict of interest.

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
