# Peer review of "Analytical Pyrolysis of the Fungal Melanins from Ochroconis spp. Isolated from Lascaux Cave, France"

_applsci, doi:10.3390/app11031198_

Round 1
Reviewer 1 Report
The paper deals with an interesting topic, anyway the discussion section must be rewritten. A deeper analysis of experimental results is need, actually, this section seems more an extended litterature background.
Please clearly specify originality of this work and its usefull for scientific community
Author Response
Thank you for your comments. Regarding your opinion on the discussion section and the need of a deeper analysis of experimental results, we would like to clarify this question. The results are presented in a total of 10 pages, with 4 figures and 4 tables. The discussion section occupies a total of 8 pages. We believe this is a good balance. In addition, we added a new section: “Current challenges and future perspectives” that meet the requirements of usefulness for the scientific community and show the way for future researches. The originality of this work is described in the Abstract: lines 21-24, Introduction, lines 54-66 and in the new section.
The discussion section is not an extended literature background. To clarify the message, we divided the data obtained by categories of compounds: aliphatic hydrocarbons, fatty acids, other aliphatic compounds, proteins, carbohydrates and polysaccharides, sterols, alkylbenzenes, alkylthiophenes and alkylnaphthalenes, and oxygen-containing compounds. In each category, the compounds obtained upon pyrolysis are discussed to the light of previous reports, in order to identify their origin (e.g. lines 352-362 for aliphatic hydrocarbons; lines 368-372 and 377-381 for fatty acids; lines 382-383, 385-387, 391-396, 403-406 and 409-411 for other aliphatic compounds, etc.
The same can be extended to the subsections on “Acid hydrolysis”, “Conventional pyrolysis vs thermochemolysis”, etc. In addition, we discussed the hypothesis for melanin formation to the light of the results obtained in the pyrolysis in the subsection “The chemical structure of Ochroconis melanins”.
We believe that in this revised manuscript we have considered and replied to all the questions posed by the three referees and adapted the new manuscript to their requirements.
Reviewer 2 Report
The paper “Analytical pyrolysis of the fungal melanins from Ochroconis spp. isolated from Lascaux Cave, France” by Saiz-Jimenez et al. is a study concerning the pyrolysis products of melanin extracted from the mycelia of different strains.
Despite thermolysis of melanin is not a novel topic, this paper merits attention, as it is well written and it contains a considerable amount of experimental work. Additionally, different methods of pyrolysis are compared and discussed. My critical comments are as follows:
- This study is essentially focused on the interpretation of peaks resulting from pyrograms. Was the pyrolysis carried out in a fixed bed under a stream of inert gas? Which gas? A little more information about the experimental setup should be given.
- The authors are very cautious and honest in admitting that there is still a long way to go before obtaining a clear correlation between the compounds resulting from melanin pyrolysis and the chemical structure of a source molecule. In section 3, a short discussion concerning these aspects would be appreciated by the reader, at least pointing out the importance of mass and energy balance, fluid dynamics and chemical kinetics underlying the modelling of a pyrolysis process for a correct identification of a source molecule. For a basic mathematical modelling of a pyrolysis process, see for example:
- R. Kaur, P. Gera, M. K. Jha, T. Bhaskar “Pyrolysis kinetics and thermodynamic parameters of castor (Ricinus communis) residue using thermogravimetric analysis”, Bioresource Technology, 250, (2018), 422–428.
- A. Chiarioni, A.P. Reverberi, B. Fabiano, V.G. Dovı` “An improved model of an ASR pyrolysis reactor for energy recovery”, Energy, 31, (2006), 2460–2468.
These studies should provide hints for the aforementioned lacking discussion and they should be acknowledged in the references.
- The acronyms should be defined as soon as they are used. Acronyms should not be used in the abstract.
- Please correct some grammar errors sparse in the text.
Author Response
- This study is essentially focused on the interpretation of peaks resulting from pyrograms. Was the pyrolysis carried out in a fixed bed under a stream of inert gas? Which gas? A little more information about the experimental setup should be given.
Answer:
The M & M section is now extended and completed with more detail about the methodology used.
“For analytical pyrolysis, the extracted melanin samples (1 mg) were deposited on small crucible capsules (Eco-cup, Frontier Labs., Fukushima, Japan). Pyrolysis-gas chromatography/mass spectrometry was performed under inert He atmosphere using a double-shot micro-furnace pyrolyzer (Frontier Labs. model 2020iD). The pyrolyzer was attached to a gas chromatograph model 6890N and to a single quadrupole mass selective spectrometer model 5973N (Agilent Technologies Inc., Santa Clara, CA, USA), as described elsewhere [20,21]. Firstly, a thermal desorption was performed by introducing the sample into a preheated micro-furnace set at sub-pyrolysis temperature of 300°C. After 2 minutes the sample was moved to a cold region of the pyrolyzer and the compounds evolved directly injected into the GC/MS for analysis. After a first desorption chromatogram was acquired, the sample was again moved to the pyrolyser hot area, now pre-heated at pyrolysis temperature of 500°C, for acquiring a second pyrolysis chromatogram. For both shots, the chromatographic and spectra acquisition conditions were the same. A low polar-fused silica capillary column (Agilent J&W HP–5 ms Ultra Inert, of 30 m x 250 μm x 0.25 μm film thickness) was used for compound separation and Helium, with a controlled flow of 1 mL min-1, as carrier gas. The GC oven temperature was set at 50°C for 1 min and then increased to 100°C at 30°C min-1, from 100°C to 300°C at 10°C min-1, and then constant at 300°C for the last 10 min. The mass spectra were acquired at 70 eV ionizing energy. Compound assignments were achieved by single-ion monitoring (SIM) and by comparison with mass spectra libraries (NIST11 and Wiley7).
The thermochemolysis of the melanins with tetramethyl ammonium hydroxide (TMAH) was performed as described by Saiz-Jimenez [22]. The analysis of the thermochemolysis compounds was done using the same instrument previously described and under the same chromatographic and mass spectra acquisition conditions.”
- . In section 3, a short discussion concerning these aspects would be appreciated by the reader, at least pointing out the importance of mass and energy balance, fluid dynamics and chemical kinetics underlying the modelling of a pyrolysis process for a correct identification of a source molecule. For a basic mathematical modelling of a pyrolysis process, see for example:
- R. Kaur, P. Gera, M. K. Jha, T. Bhaskar “Pyrolysis kinetics and thermodynamic parameters of castor (Ricinus communis) residue using thermogravimetric analysis”, Bioresource Technology, 250, (2018), 422–428.
- A. Chiarioni, A.P. Reverberi, B. Fabiano, V.G. Dovı` “An improved model of an ASR pyrolysis reactor for energy recovery”, Energy, 31, (2006), 2460–2468.
These studies should provide hints for the aforementioned lacking discussion and they should be acknowledged in the references.
Answer:
The references cited deal with thermogravimetric analysis (TGA), a technique used to investigate the decomposition behavior of a substance with respect to temperature. This TGA is used to study biomass kinetics and the technique is quite different from analytical pyrolysis. In fact, in TGA a substrate is heated from room temperature to 900°C, at constant heating rates along the time, and biomass decomposes depending on the different decomposition temperature of their constituents. Typically, in a plant biomass: cellulose, hemicellulose and lignin have different decomposition temperatures. At the end of the process a char is obtained.
In, analytical pyrolysis or flash pyrolysis, the sample is heated at a determined temperature (usually 500/600°C) for seconds, and the evolved products analyzed in a gas chromatograph-mass spectrometer. In the flash pyrolysis we used, no kinetic parameters can be calculated from curves at different heating rates, as there is only a single pyrolysis temperature. However, the analysis of the identified compounds evolved from the different pyrolysis protocols used in our work permitted to differentiate the temperatures needed for decomposition of the different melanin components: polysaccharides, proteins, lipids, aromatic compounds, temperatures which are discussed in the manuscript.
Reviewer 3 Report
This study is original in a very specific manner - it shows not the advantages but the disadvantages of an analytical procedure with important impact for specific analytical goals. The study is well designed and scientifically sound.
I have only one principal question:
Could the authors suggest an alternative analytical approach for the same analytical goals? If yes, would it be possible to compare their results with the alternative approach? I understand that it is specific and time consuming task but, at least, some kind of comparison with alternative methods might be cited.
I recommend acceptance with this minor revision response.
Author Response
Could the authors suggest an alternative analytical approach for the same analytical goals? If yes, would it be possible to compare their results with the alternative approach? I understand that it is specific and time consuming task but, at least, some kind of comparison with alternative methods might be cited.
Answer:
We thank the suggestion of the Reviewer. We added a new section: “Current challenges and future perspective”, in which we discuss alternative methods to get a pure melanin or model melanins and we present our (unfortunately) negative results regarding these methods.
In addition, the reviewer must be aware about the difficulties in comparing results obtained with different techniques. There are conventional analytical techniques widely used for the detection and quantification of melanins that have been recently reviewed by Chung et al. (2019). These are mainly based in spectroscopic and absorption spectroscopy. However, the structural analysis of melanins is not an easy task and, despite its relevance in nature, the structure of melanin remains poorly understood (Nosanchuk et al., 2015). This is due to the fact that classical biophysical methods, including crystallographic techniques, cannot be applied to long-range biopolymers acid-resistant and poorly soluble or insoluble in aqueous or organic solvents such as fungal melanins. Therefore, few analytical alternative approaches to melanin structural characterization are available and Py-GC/MS have been proposed as a destructive technique providing complementary information about melanin molecular structure to other mass spectrometric techniques combining several ionization modes, mainly MALDI (Pralea et al., 2019). However, as these authors stated, “their thorough comparative evaluation of analytical data published so far on melanin analysis has proven to be a difficult task in terms of finding equivalent results, even when the same matrix was used”. These statements have been included in the manuscript.
Moreover, we emphasize the importance of a previous knowledge of melanin types using other alternative methods, and their properties in order to select a valid experimental design able to deliver reliable results and draw consistent conclusions.
Chung, S., Lim, G.J., Lee, J.Y. (2019). Quantitative analysis of melanin content in a three-dimensional melanoma cell culture. Sci Rep 9: 780. doi: 10.1038/s41598-018-37055-y
Nosanchuk, J.D., Stark, R.E., Casadevall, A. (2015). Fungal melanin: What do we know about structure? Front. Microbiol. 6: 1463. doi: 10.3389/fmicb.2015.01463
Pralea, I.E., Moldovan, R.C., Petrache, A.M. Ilies, M., Heghes, S.C., Ielciu, I., Nicoara, R., Moldovan, M., Ene, M., Radu, M., Uifalean, A., Iuga, C.A. (2019). From extraction to advanced analytical methods: The challenges of melanin analysis. Int. J. Mol. Sci. 20: 3943. doi: 10.3390/ijms20163943
Round 2
Reviewer 1 Report
the paper is now ready for publication
Author Response
Thank you very much for accepting the manuscript
Reviewer 2 Report
The authors have given a sufficient answer to a part of the criticism raised by this reviewer.
They correctly claim that a process carried out at constant temperature does not allow obtaining information on kinetic parameters, but this is only an aspect of the discussion requested in the reviewing process. Other previously suggested aspects have been overloooked or neglected. In this context, the answer to this point is partially elusive and the authors should have at least made the effort of briefly discussing the problems related to a possible modelling, which is not limited to kinetic aspects, as clearly indicated in the previous comments of this reviewer. This in order to make this work appear as something different from a sequence of peaks from a pyrogram and nothing else...
Author Response
We thank the reviewer for his comments. We have enlarged and improved the Introduction, as recommended, and clarified the Conclusions.
Regarding the statements on “Other previously suggested aspects have been overloooked or neglected” and “the authors should have at least made the effort of briefly discussing the problems related to a possible modeling”, we remind the reviewer that the use of different model melanins and the discussion on possible macromolecular models to understand their composition by pyrolysis, has been widely discussed throughout the article (lines 546 to 642, section on “The chemical structure of Ochroconis melanins”). In addition, further comments were added in the section “Current challenges and future perspectives”, lines 654-658 and 674-681. Particular references to model melanins are presented and discussed in lines 565-622.
It may be that we do not understand what the reviewer means in his comment. In that case, we would appreciate if he could specify more clearly what he expected from our article, taking into account that all attempts to find molecular markers for melanin precursors and a macromolecule model for the Ochroconis melanin, using pyrolysis of fungal melanin models have been unsuccessful, matter that was extensively discussed in the manuscript and presented in the supplementary information. Complementary data were also discussed in a previous article on these melanins using 13C NMR. This has been reported in our manuscript. In addition, data for other fungal melanins are presented (see supplemental information) and the possible Ochroconis melanin structure is discussed to the light of these data.
Regarding the comment "This in order to make this work appear as something different from a sequence of peaks from a pyrogram and nothing else ...", we remind the reviewer that the article is not only that. In fact, we have comprehensively discussed and documented the possible origin of each group of pyrolysis compounds and their relationship with a potential macrostructure and concluded that a high number of pyrolysis products were artifacts produced during the pyrolysis. This is the main message of the article. We think that this was clearly evidenced in the manuscript.